# A landscape evaluation of caffeine citrate availability and use in newborn care across five low- and middle-income countries

Osayame A. Ekhaguere[1], Olufunke Bolaji[2]*, Helen M. Nabwera[3]¤, Andrew Storey[4], Nicholas Embleton[5], Stephen Allen[3,6], Zelalem Demeke[7], Olufunke Fasawe[8], Betty Wariari[9], Mansharan Seth[10], Lutfiyya Khan[11], Herma Hema Magge[12], Oluwaseun Aladesanmi[4]

1 Department of Paediatrics, Division of Neonatology, Indiana University School of Medicine, Indianapolis, Indiana, United States of America, 2 Department of Paediatrics and Child Health, College of Medicine, Afe Babalola University, Ado-Ekiti, Nigeria and Federal Teaching Hospital, Ido-Ekiti, Nigeria, 3 Department of Clinical Sciences, Liverpool School of Tropical Medicine, Pembroke Place, Liverpool, United Kingdom, 4 Clinton Health Access Initiative, Boston, Massachusetts, United States of America, 5 Newcastle Neonatal Service, Newcastle Hospitals NHS Trust & Newcastle University, Newcastle upon Tyne, United Kingdom, 6 Department Paediatrics, Edward Francis Small Teaching Hospital, Banjul, The Gambia, 7 Clinton Health Access Initiative, Afar Region, Ethiopia, 8 Clinton Health Access Initiative, Abuja, Nigeria, 9 Clinton Health Access Initiative, Nairobi, Kenya, 10 Clinton Health Access Initiative, New Delhi, India, 11 Clinton Health Access Initiative, Johannesburg, South Africa, 12 Bill and Melinda Gates Foundation, Seattle, Washington, United States of America

¤ Current address: Centre of Excellence for Women and Child Health, Aga Khan University, Nairobi, Kenya
* olufunke.bolaji@gmail.com

## Abstract

Apnoea of prematurity (AOP) is a common complication among preterm infants (< 37 weeks gestation), globally. However, access to caffeine citrate (CC) that is a proven safe and effective treatment in high-income countries is largely unavailable in low- and-middle income countries, where most preterm infants are born. Therefore, the overall aim of this study was to describe the demand, policies, and supply factors affecting the availability and clinical use of CC in LMICs. A mixed methods approach was used to collect data from diverse settings in LMICs including Ethiopia, Kenya, Nigeria, South Africa, and India. Qualitative semi-structured interviews and focus group discussions were conducted with 107 different health care providers, and 21 policymakers and other stakeholders from industry. Additional data was collected using standard questionnaires. A thematic framework approach was used to analyze the qualitative data and descriptive statistics were used to summarize the quantitative data. The findings indicate that there is variation in in-country policies on the use of CC in the prevention and treatment of AOP and its availability across the LMICs. As a result, the knowledge and experience of using CC also varied with clinicians in Ethiopia having no experience of using it while those in India have greater knowledge and experience of using it. This, in turn, influenced the demand, and our findings show that only 29% of eligible preterm infants are receiving CC in these countries. There is an urgent need to address the multilevel barriers to accessing CC for managing AOP in Africa. These include cost, lack of national policies, and, therefore, lack of demand stemming from its clinical equivalency with

**Data Availability Statement:** All relevant data are within the paper and its Supporting information files.

**Funding:** The landscape analysis and forecasting was funded by the Bill and Melinda Gates Foundation, and supported by the Clinton Health Access Initiative. The funders had no role in the decision to publish or the preparation of the manuscript.

**Competing interests:** The authors have declared that no competing interests exist.

aminophylline. Practical ways to reduce the cost of CC in LMICs could potentially increase its availability and use.

## Introduction

Apnoea of prematurity (AOP) is a common complication among preterm infants (< 37 weeks gestation) [1]. It is defined as breathing cessation associated with hypoxia or bradycardia in a preterm infant lasting 15 seconds or more [2]. The incidence of AOP is inversely related to gestational age and birthweight–affecting 15% of preterm infants 32–33 weeks gestation, 54% between 30 to 31 weeks [3], and nearly all infants born at <29 weeks gestation or <1,000 g [4]. Severe AOP can lead to respiratory failure [5].

The methylxanthines–caffeine citrate (CC) and theophylline (available as intravenous aminophylline)–are the pharmacological options used to prevent and treat AOP. In high-income countries, CC is the mainstay treatment for AOP [6, 7]. Its preferential use is supported by evidence showing that compared to theophylline, CC has a better side effect profile, a wider therapeutic index, a longer half-life, and does not require therapeutic drug monitoring and can be administered orally [8, 9]. Furthermore, compared to placebo, CC reduces the duration of mechanical ventilation exposure, the risk of bronchopulmonary dysplasia, duration of admission, and it is cost-effective [10–13]. This compelling evidence led the World Health Organization (WHO) to include CC in its essential list of medicines for neonatal care in 2009 [14].

Africa and South Asia bear the highest preterm birth and neonatal mortality rates, globally [15]. Therefore, optimizing the coverage of cost-effective, evidence-based interventions such as CC in these settings would significantly reduce prematurity-associated mortality and morbidity. Survey data from neonatal health care providers in low- and middle-income countries (LMICs) suggest that CC is largely unavailable, and its use is limited [16, 17]. However, the demand, policies, and supply factors affecting the availability and clinical use of CC in LMICs are unknown. Therefore, this study aimed to fill this knowledge gap.

## Methods

### Study design, setting, population, sampling

We conducted a landscape evaluation involving stakeholders in Africa (Ethiopia, Kenya, Nigeria, South Africa) and South Asia (India–five states of Delhi; Bihar, Uttar Pradesh, Telangana, and Madhya Pradesh) on CC availability and use from 1 July 2022 to 31 December 2022. We used a mixed methods study design to understand the complexity of CC availability and use across these LMICs. We selected geographically and culturally diverse countries with high annual preterm births (~200,000). In each instance, the Clinton Health Access Initiative (CHAI) had an existing partnership with the country's Ministry of Health. The selection of stakeholders within each focus country was by convenience and/or purposive sampling [18, 19]. We selected health facilities providing care for preterm infants and were able to provide the data required to achieve the study's objectives. Proximity and ease of data collection were also factored into selection by research teams.

We sought and obtained permission from the MOH to visit the facilities. We obtained permission from facility heads to collect data from facility and interview health care providers and we gave an opt-out option to those who participated in the study.

A total of 107 different healthcare providers, and 21 policymakers and other stakeholders from industry were interviewed from UNICEF. We interviewed four ministry of health

stakeholders from Ethiopia, three from India, Kenya, and Nigeria, and three from South Africa, respectively. In addition, we interviewed three stakeholders from the Ethiopian Pharmaceutical Supply Services, one stakeholder from the Ethiopian Pediatric Society, one stakeholder from the Procurement Division, Bilhar, India, one stakeholder from Kenya Medical Supplies Authority, one stakeholder from the Kenya Pharmacy and Poisons Board, and one stakeholder from Mission for Essential Drugs and Supplies Kenya.

## Data collection

**Qualitative.** The research teams conducted key informant interviews and focus group discussions (FGD's) with stakeholders in newborn health. The interviews with healthcare providers sought to explore their experience using CC as a treatment for AOP. Interviews with WHO and Ministry of Health officials sought to understand current global and national health policies and CC's inclusion in the essential drug list for AOP treatment. Interview with UNICEF Supply Division gave insights into listing CC in the commodity catalogue as part of its available products.

Interviews with major drug suppliers and distributors of CC aimed to determine the current local market pricing of CC and its alternatives within and between countries and to evaluate the factors determining the end-customer price of CC. The available average end-customer price per country was used to determine the daily cost of managing AOP for aminophylline and CC. We compared the average daily cost of aminophylline and CC for both public and private hospitals in each country. The dosing regimen for CC was a loading dose of 20 mg/kg/dose and a daily maintenance dose of between 5 to 10 mg/kg/day. The dosing regimen for aminophylline was a loading dose of 6 mg/kg administered intravenously (IV), followed by a maintenance dose of 2.5 mg/kg/dose/IV administered every 8 hours [20–22]. We interviewed three stakeholders from the Ethiopian Pharmaceutical Supply Services and one stakeholder from the Ethiopian Pediatric Society.

The CHAI country-based research teams conducted the interviews and FGDs. These were done in person or virtually over video or audio teleconferencing based on the preferences of the participants. All interviews were conducted in English. The CHAI country-based teams were situated in each country of focus and had previous training and experience conducting qualitative interviews and FGDs and in qualitative data analysis. The interviews and FGDs were semi structured using a guide with a set of open-ended questions, in a set order and allowing for in-depth insights into the subject area. (S1 Data) These guides were pilot tested across the 3 countries prior to data collection.

**Quantitative.** Additional interviews were conducted using standard questionnaires which had been piloted and refined in these settings prior to being used for data collection. (S2 Data).

The research team surveyed 107 providers: 20 from Ethiopia, 18 from India, 23 from Kenya, 28 from Nigeria, and 18 from South Africa. Healthcare providers surveyed included a mix of 27 (25%) Neonatologists, 30 (28%) Paediatricians, 10 (9%) and 24 (22%) Nurses (Table 1).

Providers were from 45 private or public health facilities across the five study countries. Of these, 12 (27%) were primary or secondary public, 7 (16%) were primary or secondary private, 25 (56%) were tertiary public, and 1 (2%) tertiary private (Table 2).

## Demand forecast for caffeine citrate

A demand forecast was conducted to determine the amount of CC needed per country. Using demographic health survey data from each country, we estimated the proportion of infants who would be eligible for CC treatment. Given AOP risk can be as high as 80% in preterm

**Table 1. Distribution of health care providers by cadre who responded to the survey.**

| Country | Ethiopia | India | Kenya | Nigeria | South Africa |
|---|---|---|---|---|---|
| Neonatologists | 4 | 8 | 2 | 15 | 0 |
| Paediatricians | 7 | 6 | 6 | 13 | 4 |
| Other Doctors | 0 | 0 | 0 | 0 | 0 |
| Nurses | 9 | 4 | 8 | 0 | 3 |
| Total | 20 | 18 | 23 | 28 | 18 |

infants with birthweight ≤1500g (very low birth weight (VLBW)), we estimated that all VLBW infants met eligibility criteria for treatment with CC [4]. We limited this forecast to public facilities where government funding constraints drug availability. We applied country-specific policies and assumptions to determine the percentage of VLBW infants who received or had a missed opportunity for CC treatment. These assumptions included, availability of CC, VLBW infants born in secondary facilities will be transferred to a tertiary center capable of providing AOP treatment; some transfers will be unsuccessful and even when successful, AOP treatment will be unavailable. (S3 Data) We defined suboptimal treatment as treatment of apnoea using Aminophylline.

## Data management and analysis

All interviews were transcribed verbatim by an experienced transcriber who was not a member of the CHAI data collection team. Authors OA and AS reviewed the interview transcripts for errors. A coding framework was generated, and an emergent thematic analysis approach was used to analyze the data, to identify patterns and themes [23]. Descriptive statistics were used to summarize the quantitative data.

## Ethical approval and informed consent

There was no institutional ethical approval as this was viewed as a collaborative project embedded within the MOH in each country, where members of the CHAI work closely with health facilities and the MOH. However, CHAI received high level approval from the leadership of the Ministries of Health in each country before data was collected. In addition, individual verbal consent was obtained from each stakeholder, healthcare provider and policy maker prior to the interviews and focused group discussions.

Inclusivity in global research: Additional information regarding the ethical, cultural, and scientific considerations specific to inclusivity in global research is included in the Supporting Information (S1 Checklist).

**Table 2. Distribution of health facilities surveyed by country, facility type, and level of care.**

| Country | Level I-II Public | Level I-II Private | Level III Public | Level III Private |
|---|---|---|---|---|
| Ethiopia | 1 | 2 | 5 | 0 |
| India | 10 | 5 | 2 | 0 |
| Kenya | 1 | 0 | 7 | 1 |
| Nigeria | 0 | 0 | 5 | 0 |
| South Africa | 0 | 0 | 6 | 0 |
| Total | 12 (27%) | 7 (16%) | 25 (56%) | 1(2%) |

## Results

### Provider perception and use

All surveyed providers from India and Nigeria reported that CC was the preferred drug to treat AOP. In Ethiopia, Kenya, and South Africa, CC preference was 50%, 56%, and 33%, respectively. In India, 95% of providers had used CC in the past, while in Ethiopia, no provider had used CC. In Kenya, CC was reported to be available for use. Providers in India, South Africa, and Nigeria reported CC was intermittently available 94%, 50%, and 33% of the time, respectively. In India and South Africa, CC is purchased by their governments for public health facilities.

A healthcare provider from Ethiopia said, *"caffeine citrate is not available in our facilities. We would use it if it were available. Even though it's included in our guidelines–essential medical list (EML)–we are not procuring the drug"*.

A neonatologist from Nigeria said, *"I'm not happy whenever I prescribe aminophylline to preterms knowing caffeine citrate is the better treatment, but it is not available for parents to purchase, and I heard it's very expensive"*.

A healthcare provider at a referral hospital in Kenya said, *"Our preference is to use caffeine citrate for only symptomatic babies due to its unavailability. Patients bring aminophylline because it's more affordable. caffeine citrate is preferred but expensive"*.

A midwife from South Africa commented that *"Because it is expensive, we make it available in small quantities in our facility"*.

A physician in charge of a special neonatal care unit in a district hospital in Uttar Pradesh, India said, *"We have requested caffeine citrate but due to ordering complexities, we have been using aminophylline"*.

**Policy relating to caffeine citrate.**   Globally, WHO first published guidelines on CC use in preterm infants in 2009. At the time of this study, the CC was included in 2020 WHO Standards for improving the quality of care for small and sick newborns in health facilities.

In Nigeria, and South Africa CC was on the national guidelines for established AOP treatment. India had no national guideline. However, healthcare providers referred to guidelines used by the All-India Institute of Medical Sciences (AIIMS) [24]. Only Ethiopia and Nigeria had prophylactic CC included in their national guideline. Ethiopia, Kenya, Nigeria, and South Africa had other methylxanthines as AOP treatment in their national guidelines, but Aminophylline was delisted from the EML in 2016 (Table 3).

**Price of caffeine citrate treatment.**   Seven stringent regulatory authority approved suppliers were found to be registered among four study countries. (Table 4) They included Chiesi (Parma, Italy), Macarthys Laboratories Ltd (Romford, United Kingdom), Aurobindo (Telangana, India) and Sun Pharma (New Jersey, USA), Micro Labs (Bangalore, India). No caffeine product was registered in Ethiopia. Four suppliers were registered in India, two in Kenya,

**Table 3. Methylxanthines and national guidelines.**

| Country | Caffeine Citrate in national guideline | Aminophylline listed as alternative | Caffeine Citrate in essential medications list | Awareness that Caffeine Citrate preferred drug (%) |
|---|---|---|---|---|
| Ethiopia | N | Y | N | 50 |
| India | N | N | Y | 100 |
| Kenya | N | Y (delisted 2016) | Y | 56 |
| Nigeria | Y | Y | Y | 100 |
| South Africa | Y | Y | Y | 33 |

**Table 4. Registration of caffeine citrate by manufacturers and country.**

| Manufacturer | Country | Brand Name | SRA* Approval | India | Nigeria | Kenya | South Africa |
|---|---|---|---|---|---|---|---|
| Chiesi | Italy | Cayona | Yes | No | No | Yes | Yes |
| Cipla | India | Capnea | No | Yes | Yes | No | No |
| McCarthy Ethypharm | United Kingdom | Martindale | Yes | No | Yes | Yes | Yes |
| Abbot Merline Pharm | India | Apnicef | Information not Available | Yes | No | No | No |
| Sun Pharma | India | Cafirate | Yes | Yes | No | No | No |
| Alliance Oversea | United Kingdom | Primicaf | Information not Available | Yes (non-SRA) | No | No | No |
| Aurobindo | India | Generic | Yes | Yes | No | No | No |

*SRA: stringent regulatory authority

Nigeria, and South Africa respectively. We engaged two distributors in India, one in Kenya, six in Nigeria, and two in South Africa.

The average end-customer price for a 25mg/ml vial of aminophylline was found to be $1.30 in Ethiopia, $0.02 in India, $2 in Kenya, $0.15 in Nigeria, and $0.30 South Africa. A 2kg premature infant with AOP will require a loading dose of 6mg/kg (12 mg) and a daily maintenance dose 2.5 mg/kg every 8 hours (15 mg/day).

The average end-customer price for a 25mg/ml vial of caffeine citrate was $2.7 in India, $22 in Kenya, $24 in Nigeria, and $24.14 in South Africa. A 2Kg premature infant with AOP will require a loading dose of 20 mg/kg (40 mg) and a daily maintenance dose 5–10 mg/kg (10–20 mg). The cost of managing AOP for 7 days with either aminophylline or caffeine in public and private medical institutions is shown in Fig 1.

In all countries, patients were responsible for all or part of the payment for CC. Only in South Africa and India was there a form of price regulation for CC in the public sector. In South Africa, the manufacturer price was the same for the two wholesale distributors we surveyed. However, the final end-customer price differed by $8.5 between retailers. This final

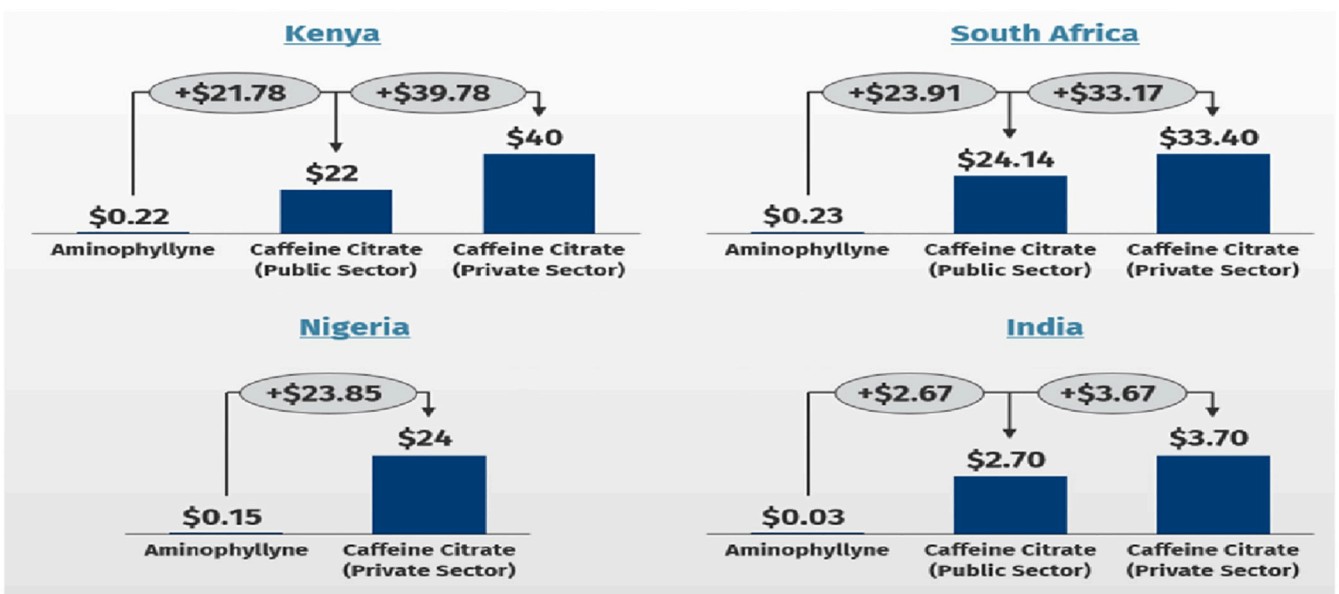

**Fig 1. Comparing the daily cost of aminophylline and caffeine citrate by health system.**

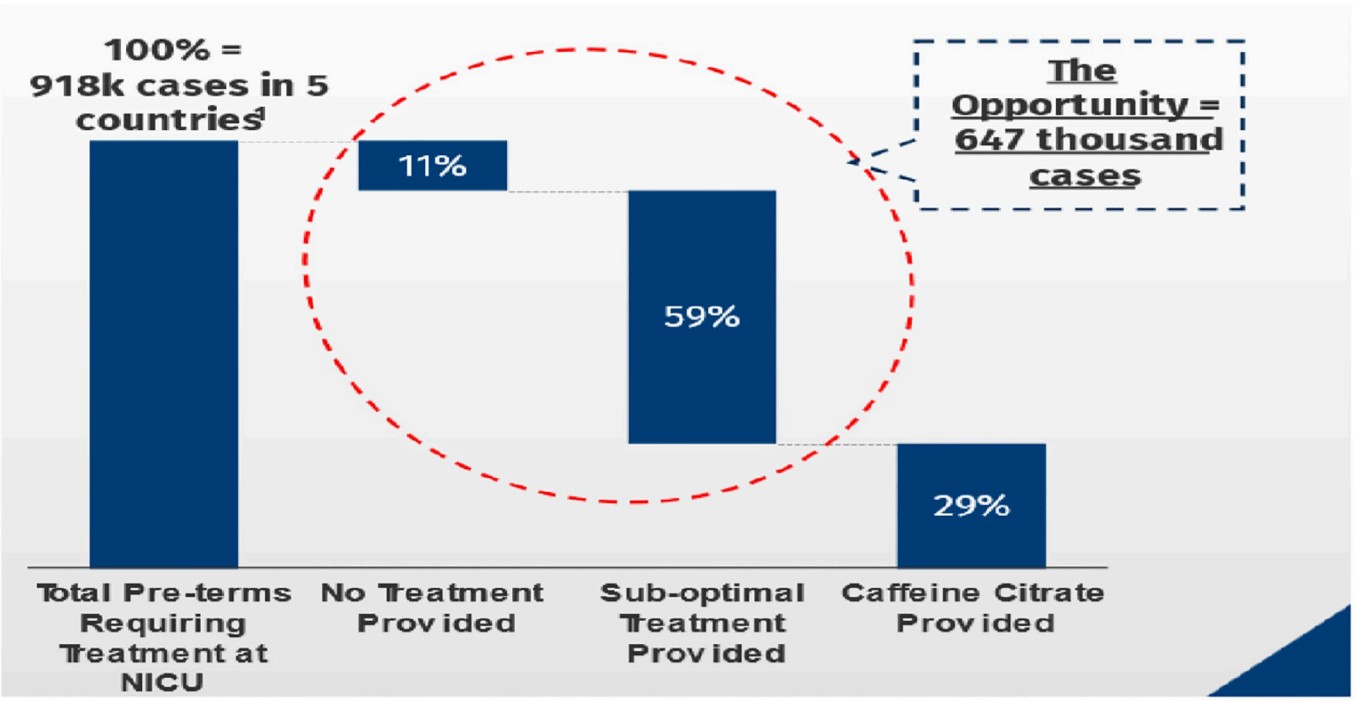

**Fig 2. Forecasting estimates on missed caffeine citrate treatment opportunities.**

price was determined by the value added taxes, logistic fees, and dispensing fees. India had two brands available–Cipla and Abbot pharmaceuticals–and there was considerable variation in price within and between brands. The Cipla brand differed by approximately $4, while the Abbott brand by approximately $5 between retailers.

**Demand forecast for caffeine citrate.** Based on the projected CC demand forecast, we estimated that out of 918,000 VLBW infants who met the criteria for AOP treatment were appropriately referred to the appropriate level of care in the 5 countries. Of the eligible premature infants who reached the appropriate level of care, 103,000 (11%), 545,000 (59%) and 270,000 (29%) received no AOP treatment, other methylxanthines, and caffeine citrate, respectively. Suboptimal treatment was defined as treatment of apnoea using Aminophylline. (Fig 2) Of the premature infants eligible for AOP treatment who reach an appropriate care facility able to provide caffeine 0% received treatment with caffeine in Ethiopia, 5% in Nigeria, 15% in Kenya, and 16% in South Africa. Only in India, at 79%, were a large proportion treated with caffeine. (Table 5).

**Table 5. Modelled out unmet need for caffeine citrate treatment per country.**

| Country | Number of Infants with AOP that received appropriate referrals to receive care | Number of referred AOP cases that did not receive AOP treatment | Number of AOP cases referred that received aminophylline (Sub-optimal treatment) | Number of AOP cases referred that received caffeine citrate |
|---|---|---|---|---|
| Ethiopia | 97,000 | 10,000 | 87,000 | 0 (0%) |
| India | 295,000 | 22,000 | 41,000 | 232,000 (79%) |
| Kenya | | | | |
| Nigeria | 394,000 | 33,000 | 342,000 | 18,000 (5%) |
| South Africa | 57,000 | 11,000 | 37,000 | 9,000 (16%) |
| Total | 918,000 | 103,000 (11%) | 545,000 (59%) | 270,000 (29%) |

## Discussion

We conducted this landscape evaluation to understand the demand, supply, and policy factors affecting CC use and availability in LMICs. 107 healthcare providers responded to the survey with most respondents being Neonatologists, Paediatricians and Nurses. Among healthcare providers in the 5 LMICs included in this study, CC preference over aminophylline was not universal. Although the most recent WHO guidance on the care of the small and sick newborns included CC [25], only Nigeria and South Africa had national guidelines for CC use for the management of AOP in preterm infants at the time of the study. All the countries except Ethiopia had CC on their essential drug list. The price of CC varied within and between countries and products, and when compared to aminophylline. Our forecast analysis estimates that about 70% of eligible patients either do not receive CC or receive aminophylline as an alternative.

Survey data from high-income countries indicate that healthcare providers prefer CC over other methylxanthines [26, 27]. Similar data on health care worker preferences and prescribing practices are lacking from LMICs. Of the 13 countries represented in an international survey among physicians in sub-Saharan Africa, only 6 countries reported consistent use of CC [16]. In our study, it was only in India and Nigeria that healthcare providers universally preferred CC over aminophylline. This may indicate a knowledge gap as there is compelling evidence predominantly from high income countries on the safety and convenience of CC over aminophylline [8, 9]. Although CC and aminophylline are equally efficacious in AOP treatment, in LMICs the side effect profile of aminophylline may not be recognized because serum and technology monitoring of levels and clinical signs of toxicity are lacking [28]. Furthermore, CC offers the added advantage of the availability of an oral formulation which is equally effective and reduces the need for intravenous access and the attendant risk of severe infection and sepsis [8, 9].

During this study, the 2020 WHO standards for improving the quality of care for small and sick newborns in health facilities included CC in its essential drugs list [29]. In the 2022 updated guideline CC is strongly recommended as first line therapy, with aminophylline as second line if CC is unavailable [30]. Only two countries had a national guideline for CC. All countries, except Ethiopia, had CC included in their essential medication list or proxy–in India–and had CC registered under their national drug regulatory body. Caffeine was added to the Ethiopian essential medication list in March of 2022.

A previous survey in Africa suggested that the cost of CC limits demand and availability [16]. We found the average end-customer price for a 20 mg/ml vial of CC ranged $2.7 in India to $24.14 in South Africa. In countries where the average family lives on less than $2 per day, no universal health insurance and drug cost is borne by the patients. The average cost of aminophylline was starkly different from CC, ranging from $0.08 in Ethiopia to $0.30 South Africa. The price difference likely contributes to its availability and demand by clinicians. The difference in the price of CC of the same product may be driven by profit, as taxes and duties were similar in the data from South Africa.

Our demand forecast shows the practice gap that exists in Africa. No eligible child in Ethiopia receives CC, compared to 78% in India. The availability of CC in India could relate to it being manufactured in India. Of the seven drug suppliers we engaged, two originate from India. While we do not have data on pricing for the Indian specific drug, it is likely that they are less expensive than imported CC. Most African countries lack drug manufacturing capabilities and a possible means to increase CC availability is to develop local drug manufacturing capability.

## Limitations

Despite our in-depth evaluation on CC use and availability in LMICs, our study had some limitations. We were unable to access end user price determinants from all CC manufacturers. We were also unable to collect detailed institutional data on CC and aminophylline use due to inadequate commodity utilization records for the products. Furthermore, the forecast data does not include estimates of patients born in lower-level facilities. Also crucially, we did not conduct any interviews with parents and parent groups of preterm infants who are key stakeholders in this process. However, we used robust demographic health survey data to estimate CC demand forecast. Finally, data for India that is a vast country, only covered five states. Nevertheless, these data provide some key insights into the diverse supply, demand, and use of CC in newborn care in LMICs that will form the basis for future research and policy agendas to address the gaps.

## Conclusion

Robust data on the safety profile, immediate and long-term beneficial effect of CC in high-income countries has made it the drug of choice for AOP treatment. However, in LMICs, especially the African region, CC use is limited. This is driven in part by cost, no national policy and perhaps a lack of demand stemming from its clinical equivalency with aminophylline. Practical ways to reduce the cost of CC in LMICs could potentially increase its availability and use. It is hoped that the findings from this paper would also facilitate research on the benefit of caffeine in the LMIC settings.

## Supporting information

**S1 Checklist. Inclusivity in global research.**
(DOCX)

**S1 Data. Qualitative data on caffeine citrate.**
(ZIP)

**S2 Data. Quantitative data on caffeine citrate.**
(ZIP)

**S3 Data. Forecast data input and assumptions for five countries.**
(DOCX)

## Acknowledgments

The authors acknowledge the country teams of Clinton Health Access Initiatives in the 5 countries where this landscape analysis was done. We also acknowledge the officials at the country Ministries of Health as well as the Pharma teams who provided information necessary for the completion of this work. Equal gratitude goes to all the health care workers in the facilities surveyed. The work done by the CHAI staff (Ryan Fu, Raj Gangadia, Devon Cain, Isa Adamu, Unathi Beku, Dr Rahel Belete, Yalewlayker Vilma, Habtamu Tezera, Dr. Carol Mwangi, Dr. Edith Mwasi, Dr. Brian Maugo, Gerald Macharia & Rosemary Kihoto) involved in data and forecast analysis is acknowledged with thanks.

## Author Contributions

**Conceptualization:** Osayame A. Ekhaguere, Olufunke Bolaji, Nicholas Embleton.

**Data curation:** Stephen Allen, Zelalem Demeke, Olufunke Fasawe, Betty Wariari, Mansharan Seth, Lutfiyya Khan.

**Formal analysis:** Osayame A. Ekhaguere, Oluwaseun Aladesanmi.

**Funding acquisition:** Zelalem Demeke, Olufunke Fasawe.

**Methodology:** Zelalem Demeke, Olufunke Fasawe, Betty Wariari, Mansharan Seth, Lutfiyya Khan, Oluwaseun Aladesanmi.

**Project administration:** Andrew Storey, Herma Hema Magge, Oluwaseun Aladesanmi.

**Resources:** Herma Hema Magge, Oluwaseun Aladesanmi.

**Software:** Herma Hema Magge.

**Supervision:** Andrew Storey, Nicholas Embleton, Stephen Allen.

**Writing – original draft:** Osayame A. Ekhaguere, Olufunke Bolaji, Helen M. Nabwera, Nicholas Embleton.

**Writing – review & editing:** Osayame A. Ekhaguere, Olufunke Bolaji, Helen M. Nabwera, Andrew Storey, Nicholas Embleton, Stephen Allen, Zelalem Demeke, Olufunke Fasawe, Betty Wariari, Mansharan Seth, Lutfiyya Khan, Herma Hema Magge, Oluwaseun Aladesanmi.

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
