## [Decision Letter · Decision Letter 0]

20 Nov 2023

PGPH-D-23-01437

A landscape evaluation of caffeine citrate availability and use in newborn care across five low- and middle-income countries

Dear Dr. Bolaji,

Thank you for submitting your manuscript to PLOS Global Public Health. After careful consideration, we feel that it has merit but does not fully meet PLOS Global Public Health’s publication criteria as it currently stands. Therefore, we invite you to submit a revised version of the manuscript that addresses the points raised during the review process.

Please note that we have only been able to secure a single reviewer to assess your manuscript. We are issuing a decision on your manuscript at this point to prevent further delays in the evaluation of your manuscript. Please be aware that the editor who handles your revised manuscript might find it necessary to invite additional reviewers to assess this work once the revised manuscript is submitted. However, we will aim to proceed on the basis of this single review if possible.

The reviewer has raised concerns regarding the small sample size of health providers interviewed in each country, the limited information regarding the specific type of healthcare providers who responded in the different sampled countries, and whether these differed across sampled regions, as this could be a major limitation in the conclusions if so.

We look forward to receiving your revised manuscript.

Kind regards,

Jennifer Tucker, PhD

Staff Editor

Journal Requirements:

1. Please include the following request in the decision letter, and ping me with follow-up. “Please include a complete copy of PLOS’ questionnaire on inclusivity in global research in your revised manuscript. Our policy for research in this area aims to improve transparency in the reporting of research performed outside of researchers’ own country or community. The policy applies to researchers who have travelled to a different country to conduct research, research with Indigenous populations or their lands, and research on cultural artefacts. The questionnaire can also be requested at the journal’s discretion for any other submissions, even if these conditions are not met.  Please find more information on the policy and a link to download a blank copy of the questionnaire here: https://journals.plos.org/globalpublichealth/s/best-practices-in-research-reporting. Please upload a completed version of your questionnaire as Supporting Information when you resubmit your manuscript.”

2. In the ethics statement in the Methods, you have specified that verbal consent was obtained. Please provide additional details regarding how this consent was documented and witnessed, and state whether this was approved by the IRB.

Additional Editor Comments (if provided):

Reviewers' comments:

Reviewer's Responses to Questions

**Comments to the Author**

1. Does this manuscript meet PLOS Global Public Health’s publication criteria? Is the manuscript technically sound, and do the data support the conclusions? The manuscript must describe methodologically and ethically rigorous research with conclusions that are appropriately drawn based on the data presented.

Reviewer #1: Yes

2. Has the statistical analysis been performed appropriately and rigorously?

Reviewer #1: N/A

3. Have the authors made all data underlying the findings in their manuscript fully available (please refer to the Data Availability Statement at the start of the manuscript PDF file)?

Reviewer #1: Yes

4. Is the manuscript presented in an intelligible fashion and written in standard English?

Reviewer #1: Yes

5. Review Comments to the Author

Reviewer #1: This is an important and well written study, with the methods and results clearly described. This paper describes important findings on the use of caffeine citrate in low income countries. Given it is one of the most commonly described drugs for neonates in high income countries with proven improvements in outcomes for very preterm infants, it is important to consider the availability of this drug in lower income settings.

Supplementary files were not provided in the file I viewed and so I was not able to review this (there was reference to a supplementary file in the main text). Also, it is not clear where the authors will make the data (such as transcripts of discussions) available.

I just have a few comments that should be addressed before publication.

One of the limitations of your qualitative analysis is the very small sample size of health providers in each county. For example, it is possible that the difference in preference for caffeine over other drugs seen across countries is due to the sample size. Please include this in your discussion.

It would also be useful to explain in the methods or results what type of health care provider answered the survey in the different countries i.e. nurse, junior doctor, consultant etc.

Were the same types of health provider surveyed in the different countries? For example it could be that the difference in preference for caffeine in the different countries could be that one country was mainly nurses who answered the survey versus another country being mainly senior doctors. Would it be possible to describe the results by categories of health care provider?

Minor comments:

page 4 - please specifiy the number of health care professionals interviewed (as well as the number of other stakeholders)

Table 2 - one cell is blank - is this not known and if so please change to not known.

Table 4 - it would be useful to include the percentage in brackets

Figure 2 - please specifiy in the figure caption what was classed as sub-optimal treatment.

Lines 219/220 don’t make sense ('Based on the projected number from the demand forecast, we estimated that out of 918,000 VLBW iinfants who met the criteria for AOP treatment received an appropriate referral for all countries.’) Please rephrase this sentence.

Please check carefully throughout for gramatical errors e.g

line 44 clinicians on Ethiopia -> clinicians in Ethiopia

line 45 The in turn -> This in turn

line 259 - CC registered in under their national drug -> delete in or rephrase

6. PLOS authors have the option to publish the peer review history of their article (what does this mean?). If published, this will include your full peer review and any attached files.

**Do you want your identity to be public for this peer review?** For information about this choice, including consent withdrawal, please see our Privacy Policy.

Reviewer #1: **Yes: **Caroline Hartley

---

## [Decision Letter · Decision Letter 1]

18 Apr 2024

PGPH-D-23-01437R1

A landscape evaluation of caffeine citrate availability and use in newborn care across five low- and middle-income countries

Dear Dr. Bolaji,

Thank you for submitting your manuscript to PLOS Global Public Health. After careful consideration, we feel that it has merit but does not fully meet PLOS Global Public Health’s publication criteria as it currently stands. Therefore, we invite you to submit a revised version of the manuscript that addresses the points raised during the review process.

Your revised manuscript was evaluated by a new reviewer who raised additional comments. Please see the comments below.

We look forward to receiving your revised manuscript.

Kind regards,

Jianhong Zhou

Staff Editor

Journal Requirements:

2. We have noticed that you have uploaded Supporting Information files, but you have not included a list of legends. Please add a full list of legends for your Supporting Information files after the references list.

Additional Editor Comments (if provided):

Reviewers' comments:

Reviewer's Responses to Questions

**Comments to the Author**

1. If the authors have adequately addressed your comments raised in a previous round of review and you feel that this manuscript is now acceptable for publication, you may indicate that here to bypass the “Comments to the Author” section, enter your conflict of interest statement in the “Confidential to Editor” section, and submit your "Accept" recommendation.

Reviewer #1: All comments have been addressed

Reviewer #2: (No Response)

2. Does this manuscript meet PLOS Global Public Health’s publication criteria? Is the manuscript technically sound, and do the data support the conclusions? The manuscript must describe methodologically and ethically rigorous research with conclusions that are appropriately drawn based on the data presented.

Reviewer #1: Yes

Reviewer #2: Yes

3. Has the statistical analysis been performed appropriately and rigorously?

Reviewer #1: Yes

Reviewer #2: Yes

4. Have the authors made all data underlying the findings in their manuscript fully available (please refer to the Data Availability Statement at the start of the manuscript PDF file)?

Reviewer #1: Yes

Reviewer #2: Yes

5. Is the manuscript presented in an intelligible fashion and written in standard English?

Reviewer #1: Yes

Reviewer #2: Yes

6. Review Comments to the Author

Reviewer #1: Thank you for addressing my comments. This paper is very important for neonatology, thank you.

Reviewer #2: This is a well written paper bringing valuable information and having potential global impact.

There are two issues that I would recommend addressing prior to publication of this paper.

The greatest benefit of caffeine citrate in high income countries is seen in extremely preterm infants born below 28 weeks of gestational age. These infants do not survive in the LMIC setting and therefore the population of infants that would benefit from caffeine citrate in the LMIC is different: above 28 weeks of GA, very preterm and moderate preterm infants.

1) There is no clear evidence that caffeine will have the same effect on when applied in the LMIC setting.

There are no published trials to show the effect of caffeine n the LMIC setting. It is

likely to have the positive effect but it would be essential to show that through the

research trial.

2) It would be important to specify throughout the paper if authors are referring to intravenous or oral caffeine preparation. If there are available data on oral preparation of caffeine availability and pricing, it would be very important to include.

Oral caffeine citrate is much cheaper than the iv preparation and it is as effective. Giving oral caffeine will reduce the need for iv access, in particular central access through the umbilical venous catheter. This is very important to minimise the risk of severe infection and sepsis and it would greatly improve survival of these infants if sepsis can be avoided.

This paper should encourage the use of oral caffeine in the LMIC setting and the main positive effect of this paper would be to facilitate further research to prove the benefit of oral caffeine in the LMIC setting in very and moderately preterm infants above 28 weeks of gestation.

7. PLOS authors have the option to publish the peer review history of their article (what does this mean?). If published, this will include your full peer review and any attached files.

**Do you want your identity to be public for this peer review?** For information about this choice, including consent withdrawal, please see our Privacy Policy.

Reviewer #1: **Yes: **Caroline Hartley

Reviewer #2: **Yes: **Sanja Zivanovic

---

## [Decision Letter · Decision Letter 2]

20 Jun 2024

A landscape evaluation of caffeine citrate availability and use in newborn care across five low- and middle-income countries

PGPH-D-23-01437R2

Dear Dr Bolaji,

We are pleased to inform you that your manuscript 'A landscape evaluation of caffeine citrate availability and use in newborn care across five low- and middle-income countries' has been provisionally accepted for publication in PLOS Global Public Health.

Best regards,

Ramachandran Thiruvengadam, M.D., Ph.D.,

Academic Editor

Reviewer Comments (if any, and for reference):

Reviewer's Responses to Questions

**Comments to the Author**

1. If the authors have adequately addressed your comments raised in a previous round of review and you feel that this manuscript is now acceptable for publication, you may indicate that here to bypass the “Comments to the Author” section, enter your conflict of interest statement in the “Confidential to Editor” section, and submit your "Accept" recommendation.

Reviewer #1: All comments have been addressed

Reviewer #2: All comments have been addressed

2. Does this manuscript meet PLOS Global Public Health’s publication criteria? Is the manuscript technically sound, and do the data support the conclusions? The manuscript must describe methodologically and ethically rigorous research with conclusions that are appropriately drawn based on the data presented.

Reviewer #1: Yes

Reviewer #2: Yes

3. Has the statistical analysis been performed appropriately and rigorously?

Reviewer #1: Yes

Reviewer #2: Yes

4. Have the authors made all data underlying the findings in their manuscript fully available (please refer to the Data Availability Statement at the start of the manuscript PDF file)?

Reviewer #1: Yes

Reviewer #2: Yes

5. Is the manuscript presented in an intelligible fashion and written in standard English?

Reviewer #1: Yes

Reviewer #2: Yes

6. Review Comments to the Author

Reviewer #1: Thank you for addressing my comments previously.

Reviewer #2: (No Response)

7. PLOS authors have the option to publish the peer review history of their article (what does this mean?). If published, this will include your full peer review and any attached files.

**Do you want your identity to be public for this peer review?** For information about this choice, including consent withdrawal, please see our Privacy Policy.

Reviewer #1: **Yes: **Caroline Hartley

Reviewer #2: **Yes: **Sanja Zivanovic
